# Leading Causes of US Deaths in the 2022

**DOI:** 10.3390/jcm13237088

**Published:** 2024-11-23

**Authors:** Camilla Mattiuzzi, Giuseppe Lippi

**Affiliations:** 1Medical Direction, Rovereto Hospital, Provincial Agency for Social and Sanitary Services (APSS) of Trento, 38068 Rovereto, Italy; camilla.mattiuzzi@apss.tn.it; 2Section of Clinical Biochemistry, University of Verona, 37134 Verona, Italy

**Keywords:** mortality, epidemiology, statistics

## Abstract

**Background and Objective:** Obtaining reliable and up-to-date information on mortality causes is essential for the planning and implementation of effective preventive interventions. We present here an analysis of the leading causes of death in the US in 2022. **Material and Methods:** We conducted an electronic search of the US Centers for Control and Prevention (CDC) Web-based Injury Statistics Query and Reporting System (WISQARS) and Wide-Ranging, Online Data for Epidemiologic Research (WONDER) to obtain information on the leading causes of death in the US for the most recent searchable year with definitive data (i.e., 2022), stratified by age and sex. **Results:** Overall, heart disease was the leading cause of death (26.2% of all deaths), followed by malignant neoplasms (22.7%), unintentional injuries (8.5%) and coronavirus disease 2019 (COVID-19; 6.9%). Although heart disease and malignant neoplasms remained the leading causes of death in both sexes, unintentional injuries were the third cause for men (10.5%), while strokes were the third cause in women (7.5%). COVID-19 remained the fourth most common cause of death in both sexes (7.1% in men and 6.8% in women). The ten most common causes of death showed an increasing mortality tendency in parallel with the aging of population, with similar trends for both sexes. The only exception was unintentional injury, which was the most common cause of death in both sexes between the ages of 15-44 years, then reached a plateau, before increasing again in people aged 65 years or older. **Conclusions:** Greater efforts should be put into prevention and education, as heart disease, cancer, and even unintentional injuries are preventable diseases.

## 1. Introduction

Accurate and current mortality statistics for the general population are essential for a range of public health, policy, and research applications. First, they can support public health planning and policy-making by helping governments and health organizations to better allocate resources in areas with higher mortality rates or where certain causes of death are more common [1]. The identification of mortality trends and causes also makes it possible to plan tailored health interventions and programs to address specific health needs and for monitoring actual effectiveness of public health interventions and efficiency of healthcare services [2]. In particular, healthcare planning and resource allocation (e.g., number and location of healthcare facilities, number of beds in hospitals and/or intensive care units, amount of medical and paramedical staff, instrumentation, funding, etc.), can be provided in areas of higher needs.

The analysis of risk factors is another important aspect that can benefit from updated mortality statistics, as these data support the investigation of causal relationships between mortality and biological, lifestyle or even environmental risk factors, paving the way for development of effective preventive measures [3]. Accurate mortality data are then essential for predicting demographic changes, which can impact economic planning, social service delivery and infrastructure development [4], but is also very useful when it comes to addressing health inequalities, especially for highlighting disparities between different populations (e.g., by age, sex, ethnicity, socioeconomic status or geographic location), thus providing useful insights for providing more equitable healthcare [5]. Finally, an accurate analysis of the leading causes of death enables the comparison between different continents, nations, and other geographic areas, which helps in reflecting global health trends and identifying health practices or behaviors associated with fewer mortality from a given disease in a certain geographical location, as well as driving clinical practice and research priorities [6].

The crisis triggered by the recent and still unresolved coronavirus disease 2019 (COVID-19) pandemic has clearly demonstrated the unpreparedness and vulnerability of most (if not all) healthcare systems globally [7,8], highlighting notable gaps in response capabilities and supply chain vulnerabilities, but also disproportionately affecting marginalized countries and/or communities, in particular frail, co-morbid or elderly patients who experienced significantly higher infection rates, more severe disease trajectories and disproportionately high mortality. This also applies to racial and ethnic minorities, who very often did not have sufficient access to preventive measures (i.e., personal protective equipment, especially face masks), treatments and vaccines [9].

It is hence clear that obtaining reliable and updated information on mortality and, especially, on the most frequent causes of death, is now unavoidable for planning and delivering effective clinical, social and even economic policies. To this end, we aim to provide here the most recent statistics on the leading causes of death in the US.

## 2. Materials and Methods

We conducted an electronic search of the US Centers for Control and Prevention (CDC) WISQARS (Web-based Injury Statistics Query and Reporting System) database [10], an interactive collection of modules containing official fatal and nonfatal statistics and injury cost data from a variety of reliable sources such as the US National Vital Statistics System (NVSS), which collects and compiles nationwide data on births, deaths, marriages, divorces and fetal deaths from vital records offices, as well as the death certificates, which contain detailed information on cause of death, demographics (e.g., sex, age, ethnicity), and geographic location. The causes of death are coded using the International Classification of Diseases 10th Revision (ICD-10), a standardized coding system published by the World Health Organization (WHO). All data are systematically reviewed by the National Center for Health Statistics (NCHS) to ensure their consistency and accuracy. The user can access the WISQARS online resource for making specific queries according to age, sex, ethnicity and geographical location [10]. The database is updated regularly, and our analysis is based on data from 2022, which represents the most recent definitive statistics on US mortality available in this database. The main purpose of the database is supporting epidemiological and clinical research aimed at investigating and attenuating the leading causes of death [10].

We specifically searched the leading causes of death in the US, using data filters as follows: data years: “2022 (i.e., the last searchable year with definitive data); number of causes: “20 intent of injury” AND “all deaths with drilldown to ICD codes”; Geography: “United States”; Age group format: “1–14 in 5-year groups; 15–65+ in 10-year groups”; Sex: “Both” AND “Males” AND “Females”; Race: “All races ethnicity”. The data were downloaded as a CSV (comma-separated value) file, converted into columns and finally transferred to an Excel spreadsheet, where they were analyzed graphically. The age-adjusted death rate was instead retrieved from the alternative CDC WONDER (Wide-Ranging, Online Data for Epidemiologic Research) database, using the same approach. This study was exempt from institutional review board review because CDC WISQARS and WONDER are anonymized and publicly accessible databases.

## 3. Results

The results of our study, which are limited to the ten most common causes of death in 2022, are summarized in Figure 1. Heart disease was the leading cause of death in the US, accounting for an average of 702,880 deaths (26.2% of all deaths), followed by malignant neoplasms (608,371 deaths; 22.7%), unintentional injuries (227,039 deaths; 8.5%) and COVID-19 (186,552 deaths; 6.9%). We found some differences between men and women; although heart disease (26.8% of all deaths for men and 25.4% for women) and malignant neoplasms (22.1% of all deaths for men and 23.3% for women) remained the leading causes of death for both sexes, unintentional injuries were the third leading cause of death for men (10.5% of all deaths), while strokes were the third leading cause of death for women (7.5% of all deaths). Notably, COVID-19 was still the fourth leading cause of death for both sexes (7.1% for men and 6.8% for women). The only sex differences in the list of the 10 leading causes of death were suicide (8th cause of death in men) and hypertension (10th cause of death in women).

The cumulative and sex-specific age distribution of the number of deaths in the US in 2022 is shown in Figure 1. The ten leading causes of death show an increasing trend parallel to the aging of the population, which is similar for both males (Figure 2) and females (Figure 3). The only exception was mortality for unintentional injuries, whose curve exhibited a sudden increase after the age of 10 years and became the leading cause of death between the ages of 15 and 44 years for both sexes, then reached a plateau followed by a further increase in those aged 65 years or older. An almost identical trend was observed for male suicides, with a peak between the ages of 35 and 44 years, followed by a plateau that continued into the older age cohort. In both sexes, mortality from Alzheimer’s disease became relevant after the age of 60 years and continued to rise thereafter, becoming the 5th causes of death in very old women and the ninth causes of death in very old men.

## 4. Discussion

There is no question that analyzing mortality data is a fundamental tool for improving public health, driving policy decisions, advancing scientific research, enabling social justice, and ultimately improving population well-being [11]. The purpose of our analysis is exactly in line with these concepts, as we aim to provide here the most up-to-date data on mortality in the US, one of the most important and representative countries in the world. The main strength of our study lies in the source of information, as the mortality data generated by our search using the WISQARS Leading Causes of Death Visualization Tool are obtained from the NVSS and death certificates, standardized using ICD-10 coding, and further processed by the NCHS. Taken together, this process provides reliable, up-to-date information on mortality trends and causes of death that can be useful for supporting a variety of public health, research and policy-making activities.

As summarized in Table 1, heart disease and cancer remained the leading causes of death in the US in 2022, echoing previously published data for 2015–2020 [12], when the leading causes of death were also heart disease, cancer, unintentional injuries and stroke, with the exception of the year 2020, when COVID-19 became the third leading cause of death in the US. Thus, four years into the COVID-19 pandemic, which has nearly revolutionized healthcare and society [13], the epidemiology of the cause of deaths in the US has remained nearly stable.

However, if we compare the number of deaths from heart disease and cancer between the years 2022 and 2020 as officially reported by the National Vital Statistics Reports [14], a modest increase can be noted for both (heart disease: 702,880 vs. 696,962, +0.8%; cancer: 608,371 vs. 602,350, +1.0%), while the mortality for COVID-19 has fallen by nearly half (350,831 vs. 186,552, −46.8%), surpassed in the 2022 list by unintentional injuries, whose mortality also increased remarkably between the years 2022 and 2020 (227,039 vs. 200,955, +13.0%). Regarding specifically heart disease, the modest but not meaningless increase can be attributed at least in part to the impact of the COVID-19 pandemic, as severe acute respiratory syndrome coronavirus 2 (SARS-CoV-2) infection is per se a trigger of myocardial ischemia and thrombosis [15]. The increase in cancer mortality is also a predictable phenomenon of the pandemic, as this has been accompanied by significant delays in cancer screening and cancer-related care [16]. As concerns unintentional injuries, our data are consistent with those recently presented in a systematic literature review that included 189 articles from many countries around the world [17], showing higher incidence of injuries/trauma (especially in men) during the pandemic. Notably, the increase relative to 2020 data can also be explained by a possible bias due to avoidance of referrals to healthcare facilities for overcrowding of emergency rooms and/or fear of infection during the initial period of the pandemic [18]. The trend of unintentional injuries as causes of death exhibits a notable pattern, with a first peak occurring between the ages of 15 and 44 years, followed by a plateau, and then increasing again in individuals aged 65 years or older. This trend can be explained by the prevalence of motor vehicle accidents, drowning, and accidental poisoning among children and young adults, whereas older adults are particularly vulnerable to falls, which are the leading cause of injury-related deaths in this age group.

As then concerns the other causes of death, the mortality for stroke has recorded a slight increase between 2022 and 2020 (165,393 vs. 160,264, +3.2%), which can also be attributed to the greater prevalence of COVID-19 cases and the direct biological impact of SARS-CoV-2 on vascular ischemic disease [19]. In contrast to the trend observed for the previous diseases, the 2022-2020 variation in mortality for chronic lower respiratory diseases was negative (147,382 vs. 152,657, −3.5%), as was that for Alzheimer’s disease (120,122 vs. 134,242, −10.5%), although the figures for the latter case may be biased by the deaths of several older people in the first two years of the pandemic, which may have then contributed to reduce the overall mortality for Alzheimer’s disease and other forms of dementia in 2022 [20]. Interestingly, the number of deaths for diabetes mellitus slightly decreased comparing the years 2022 and 2020 (101,209 vs. 102,188, −1.0%). As the prevalence of diabetes has tended to increase during the pandemic [21], the reduced mortality is not really surprising, as it can take several years for diabetic complications to develop and become potentially fatal. It may therefore take years before the effects of an increased number of diabetes diagnoses recorded during the pandemic will translate into an increased diabetes-related mortality.

It is also worth noting that the cumulative number of suicides, which ranks 11th on the list of leading causes of death in the US in both 2022 and 2020, has also considerably increased in the last two years (i.e., 49,476 vs. 45,979, +7.6%). This is also plausible given that a kaleidoscope of psychiatric illnesses (including depression) has increased significantly over the past three years for a variety of reasons, including direct biological injury to the brain from the virus, fear of disease, and the inevitable impact of societal restrictions that have severely limited personal freedom [22].

The WONDER database offers indeed an alternative source for obtaining more recent mortality data in the United States. However, the CDC cautions that “provisional data may be incomplete and may not include all deaths that occurred during a given time period, particularly for the most recent periods”. Despite these limitations, a recent study [23] utilizing provisional data has shown that heart disease and cancer remain the leading causes of death in the United States in 2023 (consistent with our analysis for 2022). Following these, unintentional injury, stroke, chronic lower respiratory disease, and Alzheimer’s disease were identified as major causes, while COVID-19 has dropped to the 10th place in the list.

A limitation of this study is that the WISQARS database provides does not support specific queries related to social determinants of health, such as education access and quality, economic stability, marital status, healthcare access and quality or social and community context. Therefore, we cannot definitively exclude the possibility that these variables may have varying impacts on the risk of mortality from specific diseases.

## 5. Conclusions

The results of our analysis confirm that heart disease, cancer, and unintentional injuries have remained the leading causes of death in the US in recent times. Therefore, reinforced efforts should be put into prevention and education, as heart disease, cancer and even unintentional injuries could be prevented by becoming aware of the risk of dying from these conditions and implementing effective lifestyle changes specifically focused on reducing the risk factors associated with these pathologies.

## Figures and Tables

**Figure 1 jcm-13-07088-f001:**
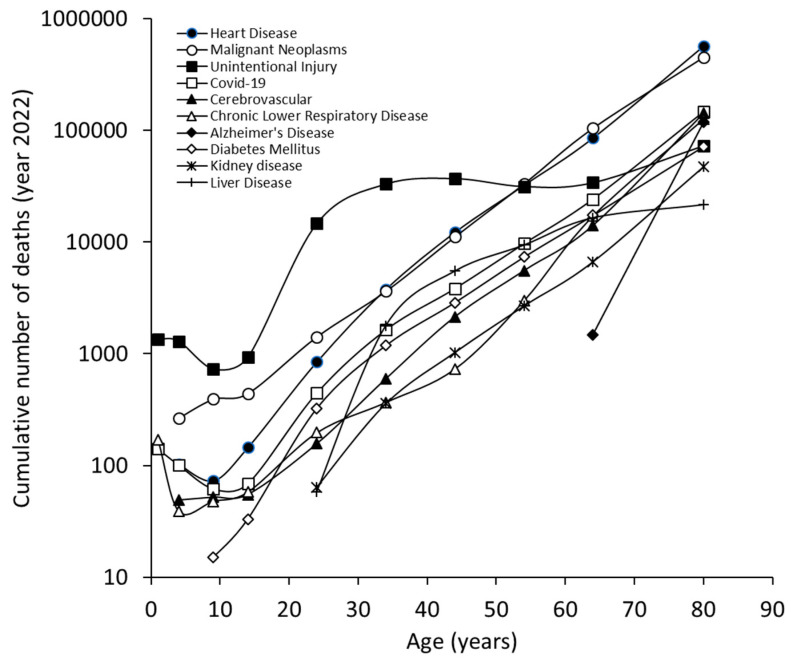
Leading causes of death in the US, year 2022, obtained from the US Centers for Control and Prevention (CDC) WISQARS (Web-based Injury Statistics Query and Reporting System) database. Cumulative number.

**Figure 2 jcm-13-07088-f002:**
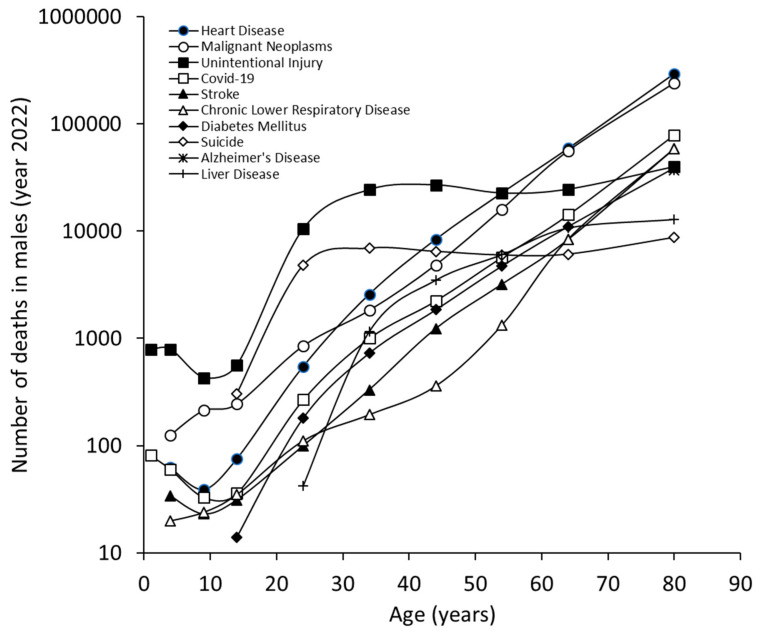
Leading causes of death in the US, year 2022, in males, obtained from the US Centers for Control and Prevention (CDC) WISQARS (Web-based Injury Statistics Query and Reporting System) database.

**Figure 3 jcm-13-07088-f003:**
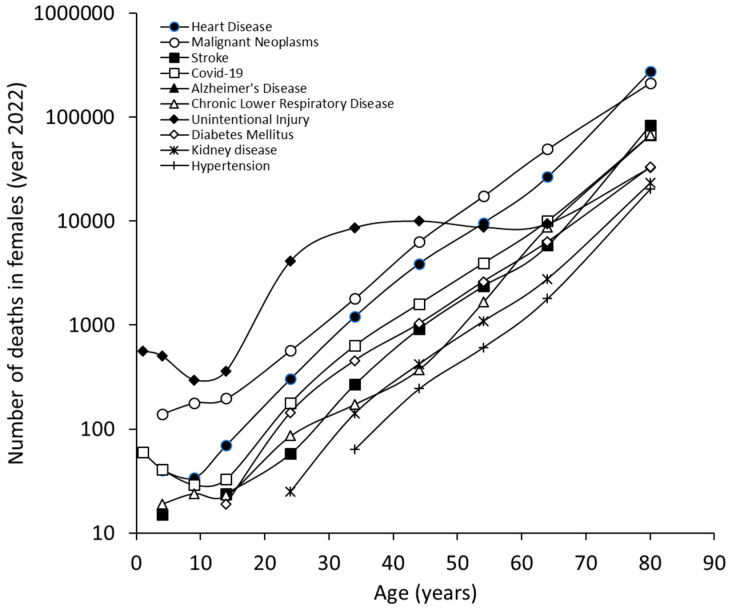
Leading causes of death in the US, year 2022, in females, obtained from the US Centers for Control and Prevention (CDC) WISQARS (Web-based Injury Statistics Query and Reporting System) database. Cumulative number.

**Table 1 jcm-13-07088-t001:** Leading causes of death in the US, year 2022, obtained from the US Centers for Control and Prevention (CDC) WISQARS (Web-based Injury Statistics Query and Reporting System) and Wonder (Wide-Ranging, Online Data for Epidemiologic Research) databases.

n.	Cause	Number of Deaths	Crude Rate	Age-Adjusted Rate	Percentage of All Deaths
**Cumulative**					
1	Heart Disease	702,880	210.9	167.2	26.2%
2	Malignant Neoplasms	608,371	182.5	142.3	22.7%
3	Unintentional Injury	227,039	68.1	64.0	8.5%
4	COVID-19	186,552	56.0	44.5	6.9%
5	Stroke	165,393	49.6	39.5	6.2%
6	Chronic Lower Respiratory Disease	147,382	44.2	34.3	5.5%
7	Alzheimer’s Disease	120,122	36.0	28.9	4.5%
8	Diabetes Mellitus	101,209	30.4	24.1	3.8%
9	Kidney disease	57,937	17.4	13.8	2.2%
10	Liver Disease	54,803	16.4	13.8	2.0%
**Males**					
1	Heart Disease	386,766	234.0	213.5	26.8%
2	Malignant Neoplasms	319,336	193.2	167.2	22.1%
3	Unintentional Injury	151,629	91.7	89.5	10.5%
4	COVID-19	102,660	62.1	56.7	7.1%
5	Stroke	71,819	43.5	40.5	5.0%
6	Chronic Lower Respiratory Disease	69,004	41.7	37.0	4.8%
7	Diabetes Mellitus	57,557	34.8	30.5	4.0%
8	Suicide	39,273	23.8	23.0	2.7%
9	Alzheimer’s Disease	37,475	22.7	23.0	2.6%
10	Liver Disease	34,340	20.8	18.0	2.4%
**Females**					
1	Heart Disease	316,114	188.2	129.5	25.4%
2	Malignant Neoplasms	289,035	172.0	124.2	23.3%
3	Stroke	93,574	55.7	38.2	7.5%
4	COVID-19	83,892	49.9	35.4	6.8%
5	Alzheimer’s Disease	82,647	49.2	32.6	6.7%
6	Chronic Lower Respiratory Disease	78,378	46.7	32.3	6.3%
7	Unintentional Injury	75,410	44.9	39.4	6.1%
8	Diabetes Mellitus	43,652	26.0	18.8	3.5%
9	Kidney disease	27,759	16.5	11.6	2.2%
10	Hypertension	23,056	13.7	9.4	1.9%

## Data Availability

Data will be available from the corresponding author upon reasonable request.

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
