# Peer review of "Leading Causes of US Deaths in the 2022"

_jcm, 2024, doi:10.3390/jcm13237088_

Round 1
Reviewer 1 Report
Comments and Suggestions for Authors
This paper is written very well. The topic is interesting and important.
Minor revision is necessary to address several concerns.
(1) This paper is relatively short, I suggest to add some discussion on the findings, such as the finding "... unintentional injury, the most common cause of death in both sexes between the ages of 15 and 44 years, then reached a plateau, for increasing again in people aged 65 years or older." Why this trend happends in US?
(2) This paper claims that "The analysis of risk factors is another important aspect that can benefit from updated mortality statistics ..." in the Introduction. However, this paper itself presents very limited contents on risk-factor analysis, only confine to age and sex. How about the other factors such as marriages, divorces, ethnicity, and geographic location ?
(3) This paper is lack of discussion on the research limitation.
Author Response
(1) This paper is relatively short, I suggest to add some discussion on the findings, such as the finding "... unintentional injury, the most common cause of death in both sexes between the ages of 15 and 44 years, then reached a plateau, for increasing again in people aged 65 years or older." Why this trend happends in US?
ANSWER: This is a good point, thanks. We have hence strengthened the discussion as mentioned below: “The trend of unintentional injuries as causes of death exhibits a notable pattern, with a first peak occurring between the ages of 15 and 44 years, followed by a plateau, and then increasing again in individuals aged 65 years and older. This trend can be explained by the prevalence of motor vehicle accidents, drowning, and accidental poisoning among children and young adults, whereas older adults are particularly vulnerable to falls, which are the leading cause of injury-related deaths in this age group.”. The article has then been lengthened by following your further suggestions as well as those of the other referees.
(2) This paper claims that "The analysis of risk factors is another important aspect that can benefit from updated mortality statistics ..." in the Introduction. However, this paper itself presents very limited contents on risk-factor analysis, only confine to age and sex. How about the other factors such as marriages, divorces, ethnicity, and geographic location ?
ANSWER: This is an interesting aspect. Unfortunately, however, WISQARS only provide data about age and sex, but not marriage and divorces. As concern geographical location, we fear that providing this information would only be of limited interest for the US readers, while we prefer to provide a more general analysis on mortality causes. Nevertheless, if the referee consider that this is a necessary information, we can make another query in the database upon a further revision (for now has just been added as a limitation).
(3) This paper is lack of discussion on the research limitation.
ANSWER: Good point, thanks. Limitation now included, as follows: “A limitation of this study was our inability to report age-adjusted mortality rates for various causes of death in the United States. This was because the WISQARS database provides information solely on the 'Number of deaths,' 'Percentage,' and 'Crude rate' [10]. the database does not support specific queries related to social determinants of health, such as education access and quality, economic stability, marital status, healthcare access and quality, or social and community context. Therefore, we cannot definitively exclude the possibility that these variables may have varying impacts on the risk of mortality from specific diseases”. If the referee has further suggestion on including additional limitation, we will be more than happy to do so.
Reviewer 2 Report
Comments and Suggestions for Authors
Dear Author, the paper is methodologically correct and has a lot of informative interest.
However, what you call the strength of the study may be a limitation.
What you have done is a summary of causes of death that can be obtained directly from the data source.
I also suggest that the rates, in addition to being crude, be calculated adjusted for age.
Author Response
Dear Author, the paper is methodologically correct and has a lot of informative interest. However, what you call the strength of the study may be a limitation. What you have done is a summary of causes of death that can be obtained directly from the data source. I also suggest that the rates, in addition to being crude, be calculated adjusted for age.
ANSWER: Thanks for this comment. However, the only information that can be obtained from WISQARS concerns: “Number of deaths”, “Percentage” and “Crude rate”. We definitely agree that providing also the age-adjusted rate would have represented an extra value for our analysis, but this is currently unfeasible. We are really sorry for not being able to overcome this technical limitation of WISQARS. This comment has been included in the discussion “A limitation of this study was our inability to report age-adjusted mortality rates for various causes of death in the United States. This was because the WISQARS database provides information solely on the 'Number of deaths,' 'Percentage,' and 'Crude rate' [10]”.
Reviewer 3 Report
Comments and Suggestions for Authors
The authors present leading causes of death for 2022 for the United States. They use CDC’s WISQARS tool to access data from the US National Vital Statistics System. Overall, the authors do a nice job presenting the data. However, I have 2 concerns.
1. The authors state that their goal is to provide the most recent and up-do-date data on leading causes of death. However, WISQARS is not the proper source for the most recent NVSS mortality data. Provisional NVSS mortality data on leading causes for 2023 and 2024 are available through CDC WONDER (https://wonder.cdc.gov/). Provisional data in CDC WONDER are updated on a weekly basis and constitute the most up-to-date figures. Also the authors seem to have missed a relevant article on leading causes in the US that includes provisional 2023 data (see https://jamanetwork.com/journals/jama/fullarticle/2822207).
2. Numbers presented in the discussion section for 2020 are based on provisional data from reference #12. These data are out of date. The authors should use final data for 2020, which can be obtained either from WISQARS or from WONDER, or from NCHS’ annual report on leading causes (https://www.cdc.gov/nchs/data/nvsr/nvsr72/nvsr72-13.pdf).
Author Response
- The authors state that their goal is to provide the most recent and up-do-date data on leading causes of death. However, WISQARS is not the proper source for the most recent NVSS mortality data. Provisional NVSS mortality data on leading causes for 2023 and 2024 are available through CDC WONDER (https://wonder.cdc.gov/). Provisional data in CDC WONDER are updated on a weekly basis and constitute the most up-to-date figures. Also the authors seem to have missed a relevant article on leading causes in the US that includes provisional 2023 data (see https://jamanetwork.com/journals/jama/fullarticle/2822207).
ANSWER: We are really thankful for this comment. Yes, we are aware of WONDER, that we have used for other analyses in the past. However, as the referee has appropriately highlighted, WONDER can only provide “provisional” data. We report here the information provided by the CDC, which makes 2023 and especially 2024 data still unreliable: “The provisional deaths are based on a current flow of mortality data in the National Vital Statistics System. National provisional counts include deaths occurring within the 50 states and the District of Columbia that have been received and coded as of the date specified. It is important to note that it can take several weeks for death records to be submitted to National Center for Health Statistics (NCHS), processed, coded, and tabulated. Therefore, the provisional data may be incomplete, and will likely not include all deaths that occurred during a given time period, especially for the more recent time periods”. We have hence included a comment at the end of our manuscript, citing also the article suggested by the referee (thanks for this suggestion): “The WONDER (Wide-Ranging Online Data for Epidemiologic Research) database offers an alternative source for obtaining more recent mortality data in the United States. However, the CDC cautions that “provisional data may be incomplete and may not include all deaths that occurred during a given time period, particularly for the most recent periods”. Despite these limitations, a recent study [23] utilizing provisional data has shown that heart disease and cancer remained the leading causes of death in the United States in 2023 (consistent with our analysis for 2022). Following these, unintentional injury, stroke, chronic lower respiratory disease, and Alzheimer’s disease were identified as major causes, while COVID-19 has dropped to 10th place on the list”. An additional aspect that must be considered, is that the article of Farida B. Ahmad and colleagues, did not perform an analysis divided for sex, as we did in our paper.
- Numbers presented in the discussion section for 2020 are based on provisional data from reference #12. These data are out of date. The authors should use final data for 2020, which can be obtained either from WISQARS or from WONDER, or from NCHS’ annual report on leading causes (https://www.cdc.gov/nchs/data/nvsr/nvsr72/nvsr72-13.pdf).
ANSWER: We are happy that the referee has raised this point, as we can clearly demonstrated that “provisional” data are unreliable. All the final data from 2020 by the National Vital Statistics Reports are different from those that we earlier reported from the publication of Ahmad et al. for the 2020. This automatically reinforces our reply to the previous point. As for this specific point, we have changed the text of the article, replacing official data (and including the new reference of the National Vital Statistics Reports).
Round 2
Reviewer 2 Report
Comments and Suggestions for Authors
I agree with your comments.
Thank you very much for clarifying my doubts.
Author Response
I agree with your comments. Thank you very much for clarifying my doubts.
ANSWER: Thanks. No additional changes are needed.
Reviewer 3 Report
Comments and Suggestions for Authors
The revision does not adequately address concerns with regard to the timeliness of the NVSS mortality data used by the authors in this study. While I don’t object to an analysis of leading causes in 2022, I do object to the characterization of the 2022 data as the most recent available data.
1. The authors insist on referring to the 2022 data as the most recent information available on leading causes of death. This is clearly not the case, as reasonably complete provisional data for 2023 were recently published in JAMA. The authors assert in their response to previous concerns that the provisional mortality data is unreliable. On the contrary, for measures such as cause-of-death ranking and even death rates, provisional mortality data have been shown to be quite reliable. E.g., published provisional cause-of-death rankings for 2022 published in May of 2023 (more than 1 year ago; see https://www.cdc.gov/mmwr/volumes/72/wr/mm7218a3.htm) are exactly the same as those presented by the authors here. The authors are correct that provisional data is somewhat incomplete and not thoroughly reviewed. However, review is continuous throughout the year and the provisional mortality data updated on a weekly basis, so the reliability of the provisional data increases steadily over time. Within 4-5 months after the end of the data year, the provisional data are more than 90% complete. The upshot is that while the provisional numbers and death rates are slightly different than those derived from the finalized data, they are widely accepted as sufficient (and are widely used in the U.S.) for the purposes of public health surveillance, planning and policy making. The value of provisional mortality data for these purposes is highlighted by the authors’ statement on lines 153-155. The authors present 2022 data as indicative of the status of the cause-of-death burden 4 years into the pandemic and assert that the epidemiology has been “nearly stable”, even though more recent, reliable data are available that show a marked shift in rankings driven by the substantial drop in COVID-19 mortality from 2022-23. In their revised manuscript, the authors do mention this more recent, provisional information (lines 207-215). But, this substantively contradicts their statement (lines 153-155) earlier in the discussion, and by excluding the provisional data from the analysis, the authors essentially ignore this important shift in epidemiology in the manuscript’s conclusions.
2. The authors’ assertion in their response that CDC WONDER “can only provide provisional data” is incorrect. WONDER includes both provisional and final data. Once the mortality data for a particular year is finalized, it is posted to WONDER and included in both the provisional and final data modules. The data presented by the authors in this manuscript can be derived from either WONDER or WISQARS. However, WISQARS, while a useful tool specifically designed for analysis of injury data, is not the primary means by which NCHS disseminates NVSS mortality data. When final mortality data are released by NCHS, the update in WONDER occurs either concurrently with the data release or shortly thereafter. NCHS pushes data directly to WONDER for dissemination; whereas before WISQARS can be updated, the National Center for Injury Prevention and Control (which maintains and updates WISQARS) must obtain the NVSS data from NCHS under a data use agreement after the data are released. Thus, the update for WISQARS typically lags that for WONDER by 2-3 months (sometimes more). So, while it is fine to use WISQARS for tabulating final data on leading causes, WISQARS is not the first place to check for the most up-to-date information, even if plans are to analyze only final mortality data. WISQARS also does not provide some statistics for non-injury causes of death (see e.g., point 3 below).
3. The authors note in their revised manuscript (lines 216-218) that a limitation of the study is an “…inability to report age-adjusted mortality rates…” The reason the authors give is that WISQARS does not provide this information. As noted above, WISQARS is primarily a tool for the analysis of injury data (age-adjusted death rates are available for injury-related causes of death in the Fatal Injury Reports). CDC WONDER does provide age-adjusted rates. It is a simple matter using WONDER to tabulate leading causes of death along with number of deaths, crude rates and age-adjusted rates. Percent of total deaths does not appear in the leading cause tabulations in WONDER, but can be easily calculated by dividing the number of cause-specific deaths by total deaths for the year. Even using WISQARS to get the basic information on leading causes, the authors could easily have used WONDER to get the age-adjusted rates for these causes. The additional work to get these data is minimal and completely eliminates a limitation of the study.
4, Lines 80-82 – I did not notice this in the first review, but this sentence describes ICD-10 as “endorsed” by the WHO. ICD-10 is a WHO product, i.e., it was created by WHO. While WHO collaborates with member states, including the U.S., in the development and maintenance of the ICD, WHO retains ownership of the classification. I suggest changing the sentence to read: “…a standardized coding system published by the World Health Organization (WHO)."
Author Response
- The revision does not adequately address concerns with regard to the timeliness of the NVSS mortality data used by the authors in this study. While I don’t object to an analysis of leading causes in 2022, I do object to the characterization of the 2022 data as the most recent available data.
ANSWER: We understand the comment of the referee and we have now modified the text keeping into account this comment. We have hence eliminated the concept that the WISQARS 2022 is the last update, emphasizing that “The database is updated regularly, and our analysis is based on data from 2022, which represents the most recent definitive statistics on US mortality available in this database”. We have then left unaltered in the text entire paragraph reporting “provisional” data for 2023, as requested by the referee upon first revision.
- The authors’ assertion in their response that CDC WONDER “can only provide provisional data” is incorrect. WONDER includes both provisional and final data. Once the mortality data for a particular year is finalized, it is posted to WONDER and included in both the provisional and final data modules. The data presented by the authors in this manuscript can be derived from either WONDER or WISQARS. However, WISQARS, while a useful tool specifically designed for analysis of injury data, is not the primary means by which NCHS disseminates NVSS mortality data. When final mortality data are released by NCHS, the update in WONDER occurs either concurrently with the data release or shortly thereafter. NCHS pushes data directly to WONDER for dissemination; whereas before WISQARS can be updated, the National Center for Injury Prevention and Control (which maintains and updates WISQARS) must obtain the NVSS data from NCHS under a data use agreement after the data are released. Thus, the update for WISQARS typically lags that for WONDER by 2-3 months (sometimes more). So, while it is fine to use WISQARS for tabulating final data on leading causes, WISQARS is not the first place to check for the most up-to-date information, even if plans are to analyze only final mortality data. WISQARS also does not provide some statistics for non-injury causes of death (see e.g., point 3 below).
ANSWER: We concur and WONDER data are now also included (i.e., the age-adjusted death ratio – see reply to point 3) in the article. Predictably, the total number of deaths and crude ratio are identical between the two databases, and were hence left unaltered. To better explain our plans, during our most recent access to the WONDER database for mortality data, the most up-to-date dataset available was 'About Underlying Cause of Death, 2018–2022, Single Race.' This indicates that 2022 is the latest year for which definitive data is accessible in WONDER. Data for subsequent years are labeled as 'preliminary,' and as we noted in our previous reply, there are significant differences between preliminary and definitive data. For example, preliminary data for 2020 published in the study by Ahmad et al. (Ahmad FB, Anderson RN, The Leading Causes of Death in the U.S. for 2020. JAMA, 2021; 325(18):1829–30) reported 690,882 deaths from heart disease. In contrast, definitive data published in the National Vital Statistics Reports indicated 696,962 deaths, a difference of over 6,000. Such discrepancies are epidemiologically substantial. Consequently, as the referee has already agreed upon, we have elected not to use “preliminary” data in our analysis or any revised version of this manuscript, as the scientific reliability of this epidemiological information is foundational to credible scientific publishing. Utilizing preliminary data would not only jeopardize the robustness of our findings but also risk exposing both the authors and the journal to rebuttals or potential retraction once the definitive data become available.
- The authors note in their revised manuscript (lines 216-218) that a limitation of the study is an “…inability to report age-adjusted mortality rates…” The reason the authors give is that WISQARS does not provide this information. As noted above, WISQARS is primarily a tool for the analysis of injury data (age-adjusted death rates are available for injury-related causes of death in the Fatal Injury Reports). CDC WONDER does provide age-adjusted rates. It is a simple matter using WONDER to tabulate leading causes of death along with number of deaths, crude rates and age-adjusted rates. Percent of total deaths does not appear in the leading cause tabulations in WONDER, but can be easily calculated by dividing the number of cause-specific deaths by total deaths for the year. Even using WISQARS to get the basic information on leading causes, the authors could easily have used WONDER to get the age-adjusted rates for these causes. The additional work to get these data is minimal and completely eliminates a limitation of the study.
ANSWER: We have followed the referee's suggestion, and the age-adjusted rate for 2022 has been extracted from the WONDER database and included in Table 1. The Materials and Methods section has also been updated accordingly. We appreciate this valuable suggestion, which has helped eliminate this limitation from the limitations of our study.
4, Lines 80-82 – I did not notice this in the first review, but this sentence describes ICD-10 as “endorsed” by the WHO. ICD-10 is a WHO product, i.e., it was created by WHO. While WHO collaborates with member states, including the U.S., in the development and maintenance of the ICD, WHO retains ownership of the classification. I suggest changing the sentence to read: “…a standardized coding system published by the World Health Organization (WHO)."
ANSWER: Completely agree. Text revised accordingly.